# The Impact of Social Network Site Addiction on Depression in Chinese Medical Students: A Serial Multiple Mediator Model Involving Loneliness and Unmet Interpersonal Needs

**DOI:** 10.3390/ijerph18168614

**Published:** 2021-08-15

**Authors:** Ruijie Gong, Yinghuan Zhang, Rusi Long, Rui Zhu, Sicong Li, Xinyi Liu, Suping Wang, Yong Cai

**Affiliations:** 1School of Public Health, Shanghai Jiao Tong University School of Medicine, Shanghai 200025, China; cynthiadt0717@163.com (R.G.); zhangyinghuan0516@163.com (Y.Z.); rusi-long@sjtu.edu.cn (R.L.); lily1998225@gmail.com (R.Z.); m15026658167@163.com (S.L.); czlxy0203@126.com (X.L.); 2Department of Prevention of Acute Infectious Diseases and Immunization, Xuhui Center for Disease Control and Prevention, Shanghai 200237, China; 3Department of Discipline Planning, Shanghai Jiao Tong University School of Medicine, Shanghai 200025, China

**Keywords:** social network sites, SNS addiction, depression, loneliness, interpersonal needs

## Abstract

The use of social network sites (SNSs) is inevitable in daily life. Everyone is likely to be addicted to SNSs, especially medical students. This study is aimed to assess the degree of SNS addiction and its relation to psychosocial factors such as depression, loneliness and unmet interpersonal needs among Chinese medical students. The cross-section survey was conducted from March to May in 2018 in Shanghai Jiao Tong University School of Medicine. Of the total 1067 participants, 33.18% had an SNS addiction, 87.7% of the participants used SNSs every day during last month and 53.42% of the participants used SNSs for at least an hour per day during the last week. SNS addiction is positively related with depression both directly and indirectly. The mediating roles of loneliness and unmet interpersonal needs on the relationship between SNS addiction and depression are significant. For the well-being of medical students, efforts should be taken to prevent them from becoming addicted to SNSs.

## 1. Introduction

Nowadays, social networking sites (SNS) are becoming increasingly popular around the world. SNSs are particularly attractive to most people, especially adolescents, for they cater to the needs of safety, association, estimation and self-realization [1], while they also create virtual spaces for seeking information, forming identity [2] and entertainment [3]. SNSs allow individuals to connect whenever and wherever possible, which also makes society increasingly inter-connected. While using SNSs brings benefits to individuals and society, they can also have negative effects, such as the problem of addiction. SNS addiction refers to a phenomenon that individuals driven by a strong motivation log on to SNSs and devote so much time and effort that it impairs other social activities, studies, work, interpersonal relationships, psychological health and well-being [4]. People may prefer online communication to face-to-face conversations, especially those with some low self-esteem. It will result in a continued and increased usage, with the consequence of neglecting offline interpersonal relationships and exacerbating psychological problems, including loneliness, depression and relationship dissatisfaction [4]. In order to cope with the resultant psychological problems, individuals will continue to engage in SNSs, which leads to a vicious cycle of addiction [5].

To some extent, almost the whole population are at risk of becoming addicted to SNSs. However, according to a meta-analysis of 10 studies worldwide, the prevalence of Internet addiction among medical students was five times higher than that of the general population [6]. The most important reason is that SNSs provide a virtual space for medical students to escape from great academic stress and social contacts since they are more vulnerable to negative psychological states (e.g., depression) [6]. A cross-sectional study conducted in India found that admittedly SNSs have their advantages, but in the present generation, the shortage dominates, especially for medical college students when considering academic performance, psychical health, mental health, social connection, etc. [7]. Up to now, there has been a lack of study on the relationship between SNS addiction and medical students in China.

Although we generally believe that SNS addiction is positively related to depression, different conclusions were drawn by several researchers on the same field. O’Keeffe first discovered “Facebook depression”, which referred to developing depressive symptoms for excessive exposure to SNSs [8]. Jelenchick found no evidence supporting a relationship between SNS addition and clinical depression [9]. Moreover, when used as a mental health intervention for young people, SNSs could significantly decrease depressive symptoms [10]. In other words, SNSs are beneficial to depression, only if used appropriately. Once SNS addiction appears, the advantage will no longer exist. Thus, the interaction between SNS addiction and depression is complex. To dispose the dispute, we assume that there are both direct and indirect relations, while there are some factors mediating this indirect association.

Tracing the history of SNSs, it was developed based on the idea that people are linked with each other via six degrees of separation [11] and tried to promote society to be inter-connected [12]. When technology makes individuals closer, more and more people feel “alone together”, which is described as always being connected via technology while feeling isolated indeed [13], then it may result in more serious problems such as depression [14]. That means when becoming obsessed with contacting via SNSs, people would find that friendship seems less meaningful, and one may also consider that communicating with the desired people may be difficult under the influence of unwished friendship [15]. It would eventually cause those SNS addicts to feel alone and unsatisfied with their interpersonal needs. Given that the unmet interpersonal needs are a result of unfulfilled social competence and belonging needs [16,17], it has been reported to be bound up with loneliness, which emerges when there is a discrepancy between desired and real social relations [18], and depression.

In this study, we try to find out whether medical students’ SNS addiction is more severe and related to psychosocial factors such as depression, loneliness and unmet interpersonal needs. Then, to better understand the relationship between SNS addiction and psychosocial factors, a serial multiple mediator model involving loneliness and unmet interpersonal needs will be built to examine the association between depression and SNS addiction based on following hypotheses:

**Hypothesis** **1** **(H1).**
*SNS addiction has substantial effects on depression, loneliness and unmet interpersonal needs.*


**Hypothesis** **2** **(H2).**
*SNS addiction has both direct and indirect effects on depression, while loneliness and unmet interpersonal needs are mediators in the relationship between SNS addiction and depression.*


## 2. Materials and Methods

### 2.1. Participants

A cross-sectional survey was conducted from March to May in 2018 at Shanghai Jiao Tong University School of Medicine. Those who met the following inclusion criteria were asked to participate: (1) aged above 18 years, (2) undergraduates, (3) proficiency in Chinese.

Assuming a 30.0% prevalence of SNS addiction in medical students [19], using an alpha of 0.05 and a relative error for sampling of 0.15, we calculated a required sample size of 415. Finally, 1067 valid questionnaires were collected without missing data among 1109 enrolled students.

### 2.2. Ethics

Written informed consent was obtained from every participant who agreed to be enrolled. First, graduate advisors agreed to distribute questionnaires in class. Then, trained investigators informed the participants about the background information on the survey (i.e., study goal, study procedure, anonymous protocol and potential risks) orally before the study began. During the recruitment, the participants were free to ask any question and to withdraw. Investigators were available for the explanation of any possible doubts while participants were filling in the questionnaire and responsible for the questionnaire collection.

### 2.3. Measures

#### 2.3.1. Demographic Characteristics

Demographic characteristics including gender, enrollment time, major, frequency of SNSs usage during last month (per week) and length of SNSs usage during last week (per day) are investigated.

#### 2.3.2. Social Networking Sites Addiction Scale (SNSAS)

The Chinese Social Networking Sites Addiction Scale was designed based on the six core symptoms of SNS addiction (mood modification, salience, tolerance, conflict, withdrawal and relapse) [20]. The validity of the SNSAS was examined in the pilot survey. Eight items (i.e., I tried to spend less time on SNSs, but I failed) answered on a 5-point Likert scale ranging from 1 (“strongly disagree”) to 5 (“strongly agree”) were included in this scale. A higher total score indicates a severer SNS addiction. The Cronbach’s alpha was 0.854.

#### 2.3.3. The 9-Item Patient Health Questionnaire (PHQ-9)

Depression was measured using the 9-item Patient Health Questionnaire, a brief screening measurement that matches the diagnostic criteria in the Diagnostic and Statistical Manual of Mental Disorders, Fourth Edition [21]. The 4-point Likert scale inquired about the participants’ frequency of depressive symptoms in the past 14 days (i.e., Over the last two weeks, how often have you been bothered by: Little interest or pleasure in doing things), ranging from 1 (“never”) to 4 (“always”). A higher total score reveals a stronger sense of depression. The Cronbach’s alpha was 0.902.

#### 2.3.4. The 8-Item UCLA Loneliness Scale (ULS-8)

The 8-item UCLA Loneliness Scale is a short version of the UCLA Loneliness Scale to assess one’s degree of loneliness [22]. Item 3 (I am an outgoing person) and item 6 (I can find companionship when I want it) were revised before calculation. The 4-point Likert scale for each item ranges from 1 (“never”) to 4 (“often”). A higher total score shows a deeper feeling of loneliness. The Cronbach’s alpha was 0.800.

#### 2.3.5. The 15-Item Interpersonal Needs Questionnaire (INQ-15)

The 15-item Interpersonal Needs Questionnaire is a short scale for estimating one’s unmet interpersonal needs [18]. Participants scored each item from 1 (“strongly disagree”) to 7 (“strongly agree”). Item 7 (These days, other people care about me), item 8 (These days, I feel like I belong), item 10 (These days, I am fortunate to have many caring and supportive friends), item 13 (These days, I feel that there are people I can turn to in times of need) and item 14 (These days, I am close to other people) were revised before computing. A higher total score means greater dissatisfaction with unmet interpersonal needs. The Cronbach’s alpha was 0.898.

### 2.4. Analysis

IBM SPSS 22.0 and JASP 0.14.1.0 were used for analysis [23]. Descriptive statistics were performed using SPSS. Pearson correlation coefficient was used to test the association between variables. Mediation analysis was performed by JASP, which provides the model of total, direct and indirect effects with bootstrap confidence interval based on 1000 resamples. A *p* value of 0.05 was considered statistically significant.

## 3. Results

### 3.1. Demographic Characteristics of Participants

The results of the descriptive statistics for the demographic characteristics of the participants are presented in Table 1. Of the total participants, 33.18% (354/1067) had an SNS addiction, 87.72% of the participants used SNSs every day during the last month and 53.42% of the participants used SNSs for at least an hour per day during the last week. While gender and major did not show obvious relationships with SNS addiction, grade was significantly related to SNS addiction; compared to freshmen, juniors (Odds Ratio (OR) = 0.548, 95% Confidence Interval (CI) = 0.388–0.774) and sophomores (OR = 0.563, 95% CI = 0.420–0.755) were less likely to be addicted to SNSs. The longer the SNSs usage time is, the more likely one would be addicted (OR = 1.523, 95% CI = 1.371–1.693).

The results of descriptive statistics for SNS addiction, loneliness, unmet interpersonal needs and depression are shown in Table 2. The average score for SNS addition was 20.19 with a range from 8.00 to 40.00. The average score for loneliness, unmet interpersonal needs and depression were 16.00 (range = 8.00–32.00), 36.62 (range = 15.00–105.00) and 15.86 (range = 9.00–36.00).

### 3.2. Correlation Coefficients of Variables

The results of Pearson correlation coefficients of SNS addiction, loneliness, unmet interpersonal needs and depression showed they are positively related to each other (Table 3).

### 3.3. Serial Multiple Mediator Model

The results of the mediation of loneliness and unmet interpersonal needs between SNS addiction and depression are described in Table 4 and Figure 1. The total effect is 0.061 (*p* < 0.001), the direct effect is 0.040 (*p* < 0.001) and the total indirect effect is 0.021 (*p* < 0.001). In addition, both of the two indirect effects are statistically significant: SNS addiction→loneliness→depression and SNS addiction→unmet interpersonal needs→depression.

## 4. Discussion

The present study indicated that 33.18% of the participants had an SNS addiction. Most of the medical students used SNSs every day and half of them used them for over one hour a day. Among the demographic characteristics, grade was the only influencing factor. The mediator model was statistically significant and matched the hypothesis proposed above that SNS addiction would result in depression, both directly and indirectly, through the mediation of loneliness and unmet interpersonal needs.

Same as the previous studies conducted worldwide, the use of SNSs was pervasive among medical students [24]. One would feel more depressed as the reliance on SNSs becomes deeper [25]. Multitudinous research showed that the use of SNSs could supplement medical education [26], thus medical students were encouraged to engage in SNSs for sharing medical materials and contents [26]. However, a majority of medical students, similar to their peers, still used SNSs for personal purposes to escape from unpleasantness in real life [27]. Medical students are more susceptible to SNS addiction on account of receiving stressful and long-term education as well as suffering from economic pressure and fierce competition in life.

Consistent with our initial propose, SNS addiction has a moderate correlation with depression among Chinese medical students. Although previous research debated whether SNS usage could help relieve individuals’ depression [28], SNS addiction did show a positive effect on depression in our study, which was consistent with several prior studies [29,30]. There are a couple of possible explanations for this. First, when people become addicted to SNSs, they are likely to be involved in a comparison in which they tend to underestimate themselves [31], resulting in depressive symptoms. On the other hand, when depressed, people may also dive into SNSs to fight the negative emotions. Second, when losing access to SNSs, SNS addicts will feel anxious and undergo a sleep quality drop [32], which can also lead to depressive symptoms. Therefore, SNS addiction interacts closely but complicatedly with depression.

The study also revealed that SNS addiction could influence depression through loneliness and unmet interpersonal needs, which can be explained as followed. First, SNS users have a stronger need for communication than those who do not use SNSs [33]. If the demand cannot be fulfilled, SNS addiction will appear. When addicted to SNSs, one will gradually give up real-world relationships, feeling lonelier and more unsatisfied with interpersonal needs [34,35], and finally depression emerges. More research is needed to further find evidence on the mediating role of loneliness and unmet interpersonal needs on the relationship between SNS addiction and depression, which is now facing a shortage.

Although this study covers all the majors and grades in medical school, it is still a single-center study. To better assess the status of SNS addiction among medical students, multiple centers are needed in the future. In the current study, we use cross-sectional data to establish a serial multiple mediator model. External verification and longitudinal studies are needed to verify the validity of the model. In addition, considering the complexity of social network addiction and psychological factors, a cohort study should be used to better explain the causal relationship.

## 5. Conclusions

The use of SNSs is inevitable in daily life, especially for medical students. When they apply SNSs, risks are still high for them to become addicted. Both direct and indirect associations are illustrated between SNSs and depression, and the mediating roles of loneliness and unmet interpersonal needs are significant. For the well-being of medical students, efforts should be taken to prevent them from becoming addicted to SNSs.

## Figures and Tables

**Figure 1 ijerph-18-08614-f001:**
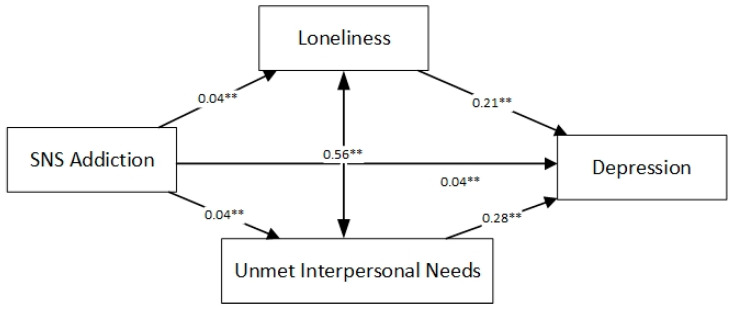
The serial multiple mediation of loneliness and unmet interpersonal needs between SNS addiction and depression. NOTE: Path coefficients were shown in standardized regression coefficients. ** *p* < 0.001.

**Table 1 ijerph-18-08614-t001:** Demographic characteristics of participants (*n*, %).

Demographic Characteristics		Number of Participants		Had SNS Addiction		OR ^a^ (95 CI% ^b^)
		*n*/*N*	Column (%)	*n*/*N*	Row (%)	
Gender ^c^						
	Male	447/1067	41.89	149/447	33.33	0.988 (0.763–1.279)
	Female	620/1067	58.11	205/620	33.06	-
Grade ^c^						
	Senior and above	18/1067	1.69	8/18	44.44	0.178 (0.016–1.980)
	Junior	203/1067	19.03	80/203	39.41	0.548 (0.388–0.774) **
	Sophomore	351/1067	32.90	136/351	38.75	0.563 (0.420–0.755) **
	Freshman	495/1067	46.39	130/495	26.26	-
Major ^c^						
	Clinical Medicine	822/1067	77.04	263/822	32.00	1.256 (0.933–1.692)
	Non-clinical Medicine ^e^	245/1067	22.96	91/245	37.14	-
Frequency of SNSs Usage During Last Month (Per Week) ^d^						
	Barely	18/1067	1.69	7/11	38.89	1.025 (0.925–1.136)
	1 day	11/1067	1.03	2/11	18.18	
	2 days	8/1067	0.75	2/8	25.00	
	3 days	9/1067	0.84	4/9	44.44	
	4 days	10/1067	0.94	1/10	10.00	
	5 days	32/1067	3.00	12/32	37.50	
	6 days	43/1067	4.03	11/43	25.58	
	7 days	936/1067	87.72	315/936	33.65	
Length of SNSs Usage During Last Week (Per Day) ^d^						
	<10 min	35/1067	3.28	8/35	22.86	1.523 (1.371–1.693) **
	10–30 min	132/1067	12.37	25/132	18.94	
	31–60 min	330/1067	30.93	76/330	23.03	
	61–120 min	307/1067	28.77	115/307	37.46	
	121–180 min	125/1067	11.72	50/125	40.00	
	>180 min	138/1067	12.93	80/138	57.97	
SNS Addiction						
	Yes	354/1067	66.82			
	No	713/1067	33.18			

^a^ Odds ratios without adjusted any confounder. ^b^ 95% confidence interval for odds ratios. ^c^ Independent variables were used as unordered categorical variables for logistic regression. ^d^ Independent variables were used as ordinal categorical variables for logistic regression. ^e^ non-clinical medicine includes clinical nursing, public health, laboratory medical science, biomedical science and medical nutrition. ** *p* < 0.001

**Table 2 ijerph-18-08614-t002:** Descriptive statistics for variables.

Measure	Mean	Median	Mode	Minimum	Maximum	Std. Deviation	Skewness	Kurtosis
SNS Addiction	20.19	20.00	16.00	8.00	40.00	5.698	0.285	0.754
Loneliness	16.00	16.00	14.00	8.00	32.00	4.403	0.299	−0.098
Unmet Interpersonal Needs	36.62	33.00	21.00	15.00	105.00	14.265	0.633	−0.121
Depression	15.86	15.00	13.00	9.00	36.00	5.269	1.333	2.342

**Table 3 ijerph-18-08614-t003:** Correlations between variables.

Variables	SNS Addiction	Loneliness	Unmet Interpersonal Needs	Depression
SNS Addiction	-	0.227 **	0.244 **	0.347 **
Loneliness	-	-	0.614 **	0.439 **
Unmet Interpersonal Needs	-	-	-	0.469 **
Depression	-	-	-	-

** *p* < 0.001.

**Table 4 ijerph-18-08614-t004:** The mediation of loneliness and interpersonal needs between SNS addiction and depression.

Path	Estimate	S.E.	95% C.I.
Total effect	0.061 **	0.005	0.048–0.072
Direct effects	0.040 **	0.005	0.027–0.052
Indirect effects			
Total indirect effects	0.021 **	0.003	0.015–0.027
Indirect 1	0.009 **	0.002	0.005–0.013
Indirect 2	0.012 **	0.002	0.008–0.017
Residual covariances	0.558 **	0.034	0.488–0.634

Indirect 1: SNS addiction→loneliness→depression. Indirect 2: SNS addiction→unmet interpersonal needs→depression. ** *p* < 0.001.

## Data Availability

The data presented in this study are available on reasonable request from the corresponding author.

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
