# Peer review of "The Impact of Social Network Site Addiction on Depression in Chinese Medical Students: A Serial Multiple Mediator Model Involving Loneliness and Unmet Interpersonal Needs"

_ijerph, 2021, doi:10.3390/ijerph18168614_

Round 1

Reviewer 1 Report

I would like to thank you for your efforts in carrying out the above research. We offer the following suggestions to improve the quality of research papers. I hope it will be helpful for research improvement.

  1.  

Please describe the ethical consent process for the research. Please record the approval number of the Ethics Review Committee.

Please provide the rationale for the sample size.

  1. Table 1.

What does OR mean? Please explain the meaning.

If it is statistically significant, please describe it.

Foot note: Please write the abbreviation on the foot note.

  1. Table 2.

Please describe the results of your research in the text.

   Describe the range, or minimun, maximum of scores for each key concept.

  1. Table 3.

Please correct the arrangement order of the description of the variable. Please place the Depression last.

Be clear on terms. (Unmet interpersonal needs, interpersonal needs )

3.3. Please describe the results of your Serial Multiple Mediator Model research.

  1. Table 4. & Fig 1.

Please describe the statistics in Table 4 and the statistics in Fig 1 without omission.

Please check the statistical results again.

Table 4. Please present the direct effcts, indirect effects, and total effects in a table respectively.

Fig 1. Please describe the term accurately. (Unmet Interpersonal Needs…..)

Author Response

Response to Reviewer 1 Comments

Point 1: Please describe the ethical consent process for the research. Please record the approval number of the Ethics Review Committee.

Response 1: It is our negligence that we did not state the ethical consent process for the research clearly and prove the approval number of the Ethics Review Committee in the first version. Our study was approved by the Ethics Committee of Shanghai Jiao Tong University School of Medicine (SJUPN-201813). And we have added that in Institutional Review Board Statement (Page 9 Line 250). The ethical consent process was described as followed in the revised manuscript in Page 3 Line 100. First, graduate advisors were agreed to distribute questionnaires in class. Second, then trained investigators were available informed the participants about the back-ground information on the survey (i.e., study goal, study procedure, anonymous protocol and potential risks) orally before the study began. During the recruitment, participants were free to ask any question and to withdraw. Investigators were available for the explanation of any possible doubts while participants were filling in the questionnaire and re-sponsible for the questionnaire collection. Thank you again for your kindly reminder.

Point 2: Please provide the rationale for the sample size.

Response 2: We are sorry that we did not report the rationale for the sample size in our first version. According to Tang[1], we assumed 30.0% prevalence of SNS addiction in medical students, using alpha of 0.05 and a relative error for sampling of 0.15, we calculated a required sample size of 415. Finally, 1067 valid questionnaires were collected without missing data among 1109 enrolled students. (Page 2 Line 92)

Your correction is very useful to us.

Reference:

  1. Tang, C.S. and Y.Y. Koh, Online social networking addiction among college students in Singapore: Comorbidity with behavioral addiction and affective disorder. Asian J Psychiatr, 2017. 25: p. 175-178.

Point 3: Table 1. What does OR mean? Please explain the meaning. If it is statistically significant, please describe it. Foot note: Please write the abbreviation on the foot note.

Response 3: Your opinion is very pertinent. It means a lot to us. We are sorry that we did not display table 1 reasonably. We optimized Table 1 in our revised version. It is worth noting that since we sorted out the data again and eliminated all the missing values. Totally, 1076 participants finished the qualified questionnaire. Since the missing values are randomly distributed, they have no great impact on the final results. Below are the results for the two analysis results. In addition, the results including statistically significant results and its meaning were described in 3.1. Demographic characteristics of participants (Page 4, Line 151). In addition, footnotes have also been updated to make the table look more understandable, for example, adding the explanation for OR, 95%CI, etc.

Table 1(V2.0). Demographic characteristics of participants (N, %).

Demographic characteristics

Number of participants

Had SNS addiction

ORa(95CI%b)

n/N

column(%)

n/N

row(%)

Genderc

Male

447/1067

41.89

149/447

33.33

0.988(0.763-1.279)

Female

620/1067

58.11

205/620

33.06

-

Gradec

Senior and above

18/1067

1.69

8/18

44.44

0.178(0.016-1.980)

Junior

203/1067

19.03

80/203

39.41

0.548(0.388-0.774)**

Sophomore

351/1067

32.90

136/351

38.75

0.563(0.420-0.755)**

Freshman

495/1067

46.39

130/495

26.26

-

Majorc

Clinical Medicine

822/1067

77.04

263/822

32.00

1.256(0.933-1.692)

Non-clinical Medicinee

245/1067

22.96

91/245

37.14

-

Frequency of SNS usage during last month (per week)d

Barely

18/1067

1.69

7/11

38.89

1.025(0.925-1.136)

1 day

11/1067

1.03

2/11

18.18

2 days

8/1067

0.75

2/8

25.00

3 days

9/1067

0.84

4/9

44.44

4 days

10/1067

0.94

1/10

10.00

5 days

32/1067

3.00

12/32

37.50

6 days

43/1067

4.03

11/43

25.58

7 days

936/1067

87.72

315/936

33.65

Length of SNS usage during last week (per day)d

<10min

35/1067

3.28

8/35

22.86

1.523(1.371-1.693)**

10-30min

132/1067

12.37

25/132

18.94

31-60min

330/1067

30.93

76/330

23.03

61-120min

307/1067

28.77

115/307

37.46

121-180min

125/1067

11.72

50/125

40.00

>180min

138/1067

12.93

80/138

57.97

SNS addiction

Yes

354/1067

66.82

No

713/1067

33.18

a Odds ratios without adjusted any confounder

b 95% confidence interval for odd ratios

c Independent variables were used as categorical variables for logistic regression

d Independent variables were used as hierarchical variables for logistic regression

e non-clinical medicine includes clinical nursing, public health, laboratory medical science, biomedical science and medical nutrition

**p<0.001

Table 1(V1.0). Demographic characteristics of participants (N, %).

Demographic characteristics

N, %

SNS addiction (N, %)

OR

Gender

Male

456(41.6%)

149(32.7%)

0.798-1.331

Female

636(58.1%)

212(33.3%)

Missing

3(0.3%)

Grade

Freshman

499(45.6%)

131(26.3%)

1.152-1.560**

Sophomore

352(32.1%)

136(38.6%)

Junior

224(20.5%)

85(37.9%)

Senior and above

18(1.6)

8(44.4%)

Missing

2(0.2)

Major

Clinical Medicine

826(75.4%)

265(32.1%)

0.903-1.106

Clinical Nursing

79(7.2%)

40(50.6%)

Public Health

35(3.2%)

11(31.4%)

Laboratory Medical Science

67(6.1%)

22(32.8%)

Biomedical Science

45(4.1%)

12(26.7%)

Medical nutrition

21(1.9%)

7(33.3%)

Missing

22(2.0%)

Frequency of SNS usage during last month (per week)

Barely

19(1.7%)

8(42.1%)

0.920-1.123

1 day

12(1.1%)

2(16.7%)

2 days

8(0.7%)

2(25.0%)

3 days

9(0.8%)

4(44.4%)

4 days

10(0.9%)

1(10.0%)

5 days

32(2.9%)

12(37.5%)

6 days

45(4.1%)

11(24.4%)

7 days

960(87.7%)

321(33.4%)

Length of SNS usage during last week (per day)

<10min

35(3.2%)

8(22.9%)

1.368-1.686**

10-30min

134(12.2%)

25(18.7%)

31-60min

340(31.1%)

78(22.9%)

61-120min

312(28.5%)

115(36.9%)

121-180min

129(11.8%)

51(39.5%)

>180min

141(12.9%)

81(57.4%)

Missing

4(0.4%)

SNS addiction

Yes

361(33.0%)

No

734(67.0%)

**P<0.001

Point 4: Table 2. Please describe the results of your research in the text. Describe the range, or minimun, maximum of scores for each key concept.

Response 4: Thank you very much for your correction. It was our negligence that did not fully describe the results in the text. The average score for SNS addition was 20.19 with a range from 8.00 to 40.00(Page 4 Line156). The average score and range for each scale was described in the text in revised manuscript and the minimum and maximum of scores for each scale was added in Table 2(Page 5 Line 170). In addition, the variable order in Table 2 was rearranged in order to be consistent with the variable order in Table 3.

Point 5: Table 3.Please correct the arrangement order of the description of the variable. Please place the Depression last. Be clear on terms. (Unmet interpersonal needs, interpersonal needs)

Response 5: Thank you for correcting our shortcomings. We have rearranged the variable order in order to make it more reasonable and corrected the terms in Table 3(Page 6 Line 175).

Point 6: Please describe the results of your Serial Multiple Mediator Model research.

Response 6: Thank for your reminder. The results of serial multiple mediator model were described in the revised manuscript that the total effect is 0.061(p<0.001), the direct effect is 0.040(p<0.01) and the total indirect effect is 0.021(p<0.001). In addition, both two indirect effects are statistically significant: SNS addiction → loneliness → depression and SNS addiction → unmet interpersonal needs → depression (Page 7 Line 179).

Point 7: Table 4. & Fig 1. Please describe the statistics in Table 4 and the statistics in Fig 1 without omission. Please check the statistical results again. Table 4. Please present the direct effects, indirect effects, and total effects in a table respectively. Fig 1. Please describe the term accurately. (Unmet Interpersonal Needs…..)

Response 7: Your correction is very useful for us that we checked our statistical results and described that in Table 4 and in Fig1. The direct effects, indirect effects, and total effects are also presented in Table 4 (Page 7 Line 184) and the term are corrected (Page 8 Line 190) in the revised manuscript.

Reviewer 2 Report

This is an interesting paper dealing with the social network site addiction in a sample of medical students in China. 

It is interesting that over 30% of the studied medial students seem addicted to social network sites.

1.However, when compared to other majors (like medical nursing, nutrition and laboratory medicine) they seem less addicted, contradicting the initial reference that medical students exceed the general population in terms of SNS addiction.I think this information from the results should be stated.

 2.There was also a problem when adding the different students in the different majors in the sense that the total of the percentages along with the missing majors exceeds 100%.(Table 1). Also on the same table concerning frequency on day 1 there is an exclamation mark in the parenthesis  of the percentages, instead of 1.

3. Also, I was not convinced by the results in Table 4 that loneliness and interpersonal needs mediate the relation between SNS addiction and depression. It is known from previous studies (  f. ex .Cacioppo JT, Hawkley LC, Thisted RA. Perceived social isolation makes me sad: 5-year cross-lagged analyses of loneliness and depressive symptomatology in the Chicago Health, Aging, and Social Relations Study. Psychol Aging. 2010 Jun;25(2):453-63) that loneliness precedes depression and that it doesn't presuppose SNS addiction to lead to depression ( evident also from the stronger correlations between depression and loneliness in Table 3).

Wouldn't it be more plausible that loneliness leads to SNS addiction and by not relieving or even by augmenting loneliness, the person ends up feeling depressed and not the other way round as hypothesised in the discussion?

4. Finally, in the abstract and throughout the text results referring to medical students could be more emphatically juxtaposed to the other majors, if  the small number of the other students in non-medical majors can build a  statistically valid control group.

Overall, an interesting paper with expectable results.

Author Response

Response to Reviewer 2 Comments

Point 1: However, when compared to other majors (like medical nursing, nutrition and laboratory medicine) they seem less addicted, contradicting the initial reference that medical students exceed the general population in terms of SNS addiction.I think this information from the results should be stated.

Response 1: Thank you for your comments, which are very useful to us. As we mentioned, due to the characteristics of medical students, they may be more likely to be addicted to SNS than ordinary population. However, we only conducted our study in Shanghai Jiao Tong University School of Medicine where there were only clinical medicine and non-clinical medicine students (including clinical nursing, public health, laboratory medical science, biomedical science and medical nutrition). So, we didn't compare the prevalence of SNS addiction of medical students with that of the general population. This is also the limitation of this study. Thus, in our research limitations, we also mentioned that multicenter research should be conducted in the future (Page 9 Line 231). And we also compared the differences between clinical medicine students and non-clinical medicine students. However, there were no statistically significant. We also stated the results in our revised manuscript (Page 4 Line 149).

Point 2: There was also a problem when adding the different students in the different majors in the sense that the total of the percentages along with the missing majors exceeds 100%.(Table 1). Also on the same table concerning frequency on day 1 there is an exclamation mark in the parenthesis of the percentages, instead of 1.

Response 2: Thank you for your correction. We are very sorry for making such mistake. In the original table 1(V1.0), we marked the percentage as the constituent ratio, which has been corrected to the rate in the revised manuscript (Table 1 [V2.0]). We sorted out the data again and eliminated all the missing values. Totally, 1076 participants finished the qualified questionnaire. Since the missing values are randomly distributed, they have no great impact on the final results. Below are the results for the two analysis results.

Table 1(V2.0). Demographic characteristics of participants (N, %).

Demographic characteristics

Number of participants

Had SNS addiction

ORa(95CI%b)

n/N

column(%)

n/N

row(%)

Genderc

Male

447/1067

41.89

149/447

33.33

0.988(0.763-1.279)

Female

620/1067

58.11

205/620

33.06

-

Gradec

Senior and above

18/1067

1.69

8/18

44.44

0.178(0.016-1.980)

Junior

203/1067

19.03

80/203

39.41

0.548(0.388-0.774)**

Sophomore

351/1067

32.90

136/351

38.75

0.563(0.420-0.755)**

Freshman

495/1067

46.39

130/495

26.26

-

Majorc

Clinical Medicine

822/1067

77.04

263/822

32.00

1.256(0.933-1.692)

Non-clinical Medicinee

245/1067

22.96

91/245

37.14

-

Frequency of SNS usage during last month (per week)d

Barely

18/1067

1.69

7/11

38.89

1.025(0.925-1.136)

1 day

11/1067

1.03

2/11

18.18

2 days

8/1067

0.75

2/8

25.00

3 days

9/1067

0.84

4/9

44.44

4 days

10/1067

0.94

1/10

10.00

5 days

32/1067

3.00

12/32

37.50

6 days

43/1067

4.03

11/43

25.58

7 days

936/1067

87.72

315/936

33.65

Length of SNS usage during last week (per day)d

<10min

35/1067

3.28

8/35

22.86

1.523(1.371-1.693)**

10-30min

132/1067

12.37

25/132

18.94

31-60min

330/1067

30.93

76/330

23.03

61-120min

307/1067

28.77

115/307

37.46

121-180min

125/1067

11.72

50/125

40.00

>180min

138/1067

12.93

80/138

57.97

SNS addiction

Yes

354/1067

66.82

No

713/1067

33.18

a Odds ratios without adjusted any confounder

b 95% confidence interval for odd ratios

c Independent variables were used as categorical variables for logistic regression

d Independent variables were used as hierarchical variables for logistic regression

e non-clinical medicine includes clinical nursing, public health, laboratory medical science, biomedical science and medical nutrition

**p<0.001

Table 1(V1.0). Demographic characteristics of participants (N, %).

Demographic characteristics

N, %

SNS addiction (N, %)

OR

Gender

Male

456(41.6%)

149(32.7%)

0.798-1.331

Female

636(58.1%)

212(33.3%)

Missing

3(0.3%)

Grade

Freshman

499(45.6%)

131(26.3%)

1.152-1.560**

Sophomore

352(32.1%)

136(38.6%)

Junior

224(20.5%)

85(37.9%)

Senior and above

18(1.6)

8(44.4%)

Missing

2(0.2)

Major

Clinical Medicine

826(75.4%)

265(32.1%)

0.903-1.106

Clinical Nursing

79(7.2%)

40(50.6%)

Public Health

35(3.2%)

11(31.4%)

Laboratory Medical Science

67(6.1%)

22(32.8%)

Biomedical Science

45(4.1%)

12(26.7%)

Medical nutrition

21(1.9%)

7(33.3%)

Missing

22(2.0%)

Frequency of SNS usage during last month (per week)

Barely

19(1.7%)

8(42.1%)

0.920-1.123

1 day

12(1.1%)

2(16.7%)

2 days

8(0.7%)

2(25.0%)

3 days

9(0.8%)

4(44.4%)

4 days

10(0.9%)

1(10.0%)

5 days

32(2.9%)

12(37.5%)

6 days

45(4.1%)

11(24.4%)

7 days

960(87.7%)

321(33.4%)

Length of SNS usage during last week (per day)

<10min

35(3.2%)

8(22.9%)

1.368-1.686**

10-30min

134(12.2%)

25(18.7%)

31-60min

340(31.1%)

78(22.9%)

61-120min

312(28.5%)

115(36.9%)

121-180min

129(11.8%)

51(39.5%)

>180min

141(12.9%)

81(57.4%)

Missing

4(0.4%)

SNS addiction

Yes

361(33.0%)

No

734(67.0%)

**P<0.001

Point 3: Also, I was not convinced by the results in Table 4 that loneliness and interpersonal needs mediate the relation between SNS addiction and depression. It is known from previous studies (  f. ex .Cacioppo JT, Hawkley LC, Thisted RA. Perceived social isolation makes me sad: 5-year cross-lagged analyses of loneliness and depressive symptomatology in the Chicago Health, Aging, and Social Relations Study. Psychol Aging. 2010 Jun;25(2):453-63) that loneliness precedes depression and that it doesn't presuppose SNS addiction to lead to depression ( evident also from the stronger correlations between depression and loneliness in Table 3).

Wouldn't it be more plausible that loneliness leads to SNS addiction and by not relieving or even by augmenting loneliness, the person ends up feeling depressed and not the other way round as hypothesised in the discussion?

Response 3: Thank you very much for your pertinent comments. The reference you provided have given us great inspiration. There was evidence that the influence of SNS may be age-related[1]. If possible, we are also very willing to do research in this direction. Second, as you pointed out, this is a very important hypothesis that loneliness would lead to SNS addiction and by not relieving or even by augmenting loneliness, the person ends up feeling depressed. That means there may be a dynamic cycle between SNS addiction and psychosocial factors. As we mentioned in limitation, it is to be regretted that the present study is a cross-section study. We could only try to find the relationship between SNS addiction and depression based on our present data but hard to explain how the factors interacted with each other (Page 9 Line 234). According to our present results, loneliness did mediate SNS addiction to depression significantly. And we would pay attention to the wording when we draw a conclusion. Thank you very much for your comments. Next, we will focus on studying the relationship between loneliness, depression and social network addiction. Perhaps establishing a cohort can explain the relationship more clearly.

Reference:

  1. Kuss, D.J. and M.D. Griffiths, Social Networking Sites and Addiction: Ten Lessons Learned. Int J Environ Res Public Health, 2017. 14(3).

Point 4: Finally, in the abstract and throughout the text results referring to medical students could be more emphatically juxtaposed to the other majors, if  the small number of the other students in non-medical majors can build a  statistically valid control group.

Response 4: Thank you very much for your suggestions. Since the sample size of some non-clinical medicine major such as medical nutrition(n=21) was quite small, we have integrated majors in the revised manuscript, and set the non-clinical medicine major as the control group, including clinical nursing, public health, laboratory medical science, biomedical science and medical nutrition. However, the present study did not find the significant difference between the two. Thank you again for your suggestions, which will be of great help to our next research.

Round 2

Reviewer 1 Report

I would like to thank you for your efforts in revisioning the manuscript. I offer the following suggestions to improve the quality of research papers. I hope it will be helpful for research improvement.

  1. Table 1.

Please check the following statistics for odds ratio and correct it.   

Ex)   Frequency of SNS usage  -----    1.025

Length of SNS usage    ------    1.523

  1. Table 4.

Please check the title of the table. 

The median  ----   (??) 

Author Response

Response to Reviewer 1 Comments

Point 1: Table 1.

Please check the following statistics for odds ratio and correct it.  

Ex)   Frequency of SNS usage  -----    1.025

Length of SNS usage    ------    1.523

Response 1: We confirmed that the odds ratio was right. We believe that these two variables (Frequency of SNS usage and Length of SNS usage) are more suitable to be defined as ordinal categorical variables rather than unordered categorical variables. Our output in SPSS can be found in the attachment. If we defined the two variables as unordered categorical variables, the results are consistent.

Point 2: Table 4.

Please check the title of the table.

The median  ----   (??)

Response 2: We are sorry that we made the spelling mistake. We have corrected the title of table 4 in the revised manuscript into “Table 4. The mediation of loneliness and interpersonal needs between SNS addiction and depression.”.
